# Wave-Powered and Zero-Discharging Membrane-Distillation Desalination System: Conceptual Design and Analysis

Gyeong Sung Kim and Yunho Hwang *

Center for Environmental Energy Engineering, Department of Mechanical Engineering, University of Maryland, College Park, MD 20742, USA; gskim@umd.edu
* Correspondence: yhhwang@umd.edu; Tel.: +1-301-405-5247

**Abstract:** There are many islands without full access to electricity around the world. These energy-poor regions generally have drinking water supply issues too. Renewable energy-powered desalination units can convert seawater to freshwater by using such as oceanic wave energy to mitigate the water limitation in small islands. A novel wave-powered floating desalination system (WavoWater) was proposed for easy on-site deployment and minimal environmental impact. WavoWater can produce freshwater using a vacuum-applied air-gap membrane distillation (AGMD) system, and the heat needed for the AGMD is provided through a heat pump powered by wave energy. Small-scale experiments were conducted to estimate the water generation rate of the vacuum-applied AGMD, and the WavoWater system modeling was developed based on the experimental results and wave data observed near the City of Newport, OR, USA. Fast Fourier transform was applied to estimate the wave energy spectrum in a random sea wave state. It was evaluated that 1 m-diameter WavoWater can produce 12.6 kg of fresh water per day with about 3.1 kWh of wave energy. With the performance evaluation, the aspects of zero discharging and minimal environmental impact were also highlighted for the stand-alone wave-powered desalination system.

**Keywords:** wave-powered desalination; heat pump; airgap membrane distillation; vacuum





## 1. Introduction

There are many islands and isolated coastal areas without reliable access to electricity worldwide [1,2]. For example, it has been reported that only around 70% of the dwellers in small islands in the Pacific Ocean could access electricity [2]. Even the residents in islands who are able to use electricity mainly depend on diesel-powered electric generators that generally show less than 40% efficiency [1,3] and high costs due to the additional fuel transportation cost from the mainland [4]. These energy-poor regions also can face water shortage issues since they are vulnerable to climate change, such as extreme weather events, sea-level rises, and temperature increases [5]. Therefore, desalination technologies have aroused interest as an alternative way to supply fresh water to islands suffering from freshwater shortages. However, most desalination plants require stable grids to operate hydraulic pumps and other auxiliary electric devices of the systems. Moreover, seawater desalination is known as an energy-intensive process for driving fluids through membrane modules, providing heat to thermally-driven processes (e.g., distillation, evaporation, and condensation), and operating mechanical components (e.g., valves, compressors, and pumps).

Renewable-energy-powered desalination units can convert seawater or brackish water to freshwater using off-grid energy, such as solar, wind, or oceanic wave energies, to mitigate this water limitation in small islands. Given this background, numerous technical concepts have been proposed to provide freshwater for off-grid regions, and most of the experiments to date have been performed focusing on utilizing solar and wind energies [6]. Renewable energy-based desalination technologies have been mainly studied for reverse

osmosis (RO), multi-stage flash distillation (MSF), multi-effect distillation (MED), and membrane distillation (MD).

Although wave-powered desalination has not been a mainstream application, it has garnered great interest and momentum recently in a bid to target off-grid and coastal communities. For example, in 1982, Delbuoy [7] developed RO membrane modules with a linear hydraulic pump driven by the heaving motion of waves to supply pressurized seawater and produced up to 2.0 m³/day of fresh water. In 2009, WaveCatcher [8] was designed to pressurize the feed water for the RO process. The researchers demonstrated 420 kPa of maximum pressure with the small-scale prototype (1:6) and reported that it might generate the maximum pressure of 6 MPa with a full-scale system. In 1987, the Edinburgh Duck [9] was invented based on a mechanical vapor compression (MVC) process while utilizing the nodding motion of a floating body and demonstrated the potential for seawater desalination with a 1:33 scale model in its following study [10].

Despite the steady progress of research on seawater desalination technology development using wave power, it is not easy to find research on wave power-based MD systems. MD can utilize low-grade waste heat to separate saline feedstock, which has emerged as an advantage compared to other desalination methods and has been mainly studied as a desalination process utilizing waste heat [11]. Therefore, in the existing desalination methods using wave power, few studies have been conducted on grafting MD, which is a desalination technology of a heat treatment method. However, considering that the seawater temperature is maintained high in water-scarce areas near the equator, such as the countries of the Middle East and North Africa or island countries in the Pacific, a wave-power-based MD desalination can be considered. Therefore, unlike previous studies, we develop a wave-power desalination system using an MD process and evaluate its performance. In particular, this study utilized the wave measurement data presented in the literature based on the assumption that the system is operated off the coast of Oregon, USA, an area in which the water shortage problem has recently been addressed. This article first describes the new desalination concept along with how each of its major components works. We then discuss a random-wave numerical model for predicting power and hydraulic power generation in systems. Next, the expected water production model was estimated through a bench-scale experiment.

## 2. Wave-Powered Desalination System

### 2.1. Concept Description

The MD system basically needs a heating process. Therefore, the wave power-based MD desalination device we propose applies a heat pump (HP) system driven by wave power. The desalination unit is designed as a single buoy, and the heat pump inside the desalination unit absorbs the thermal energy of the surrounding seawater through the evaporator and uses it as a heat source for the MD desalination method. We named this desalination device WavoWater. The HP is powered by a wave energy converter (WEC) that utilizes the heaving motion of the cylindrical body of WavoWater.

The working principles of the desalination process of WavoWater are described in Figure 1. The mechanical energy of wave motions is converted to electric power by a power take-off (PTO) device to provide electricity for the heat pump (HP) operation. Next, the hydrothermal energy of seawater is collected by the HP to heat up the draw solution (DS), which always maintains a higher concentration than the surrounding seawater body. The DS creates a natural osmotic pressure between two solutions so that water molecules of the seawater can penetrate the forward-osmosis (FO) membrane and move to the DS. Then, water vapor flux is created from the DS to the air gap by the vapor pressure difference between the two sides of the MD membrane. The MD membrane is hydrophobic so that only water vapor is allowed to pass through the MD membrane pores from the feed to the permeate side. Finally, the vapor in the air gap is condensed and collected after contacting the cooling surface provided by the HP system. The latent heat recovered during the condensation is reused for heating the DS. This working process is designed to be

continuously operated as long as the electrical power is supplied enough by the sea wave motions, and it can be described that WavoWater consists of five key processes to produce fresh water from the sea water:

(i)     converting sea-wave energy to electric power for the heat pump (HP) operation using the power take-off (PTO) inside the WavoWater;
(ii)    absorbing hydrothermal energy from the sea and heating DS using the HP;
(iii)   generating water vapor flux through the hydrophobic membrane by the evaporation process;
(iv)    condensing water vapor in the air gap; and
(v)     transporting water molecules from the sea to the DS by the natural osmotic phenomenon.

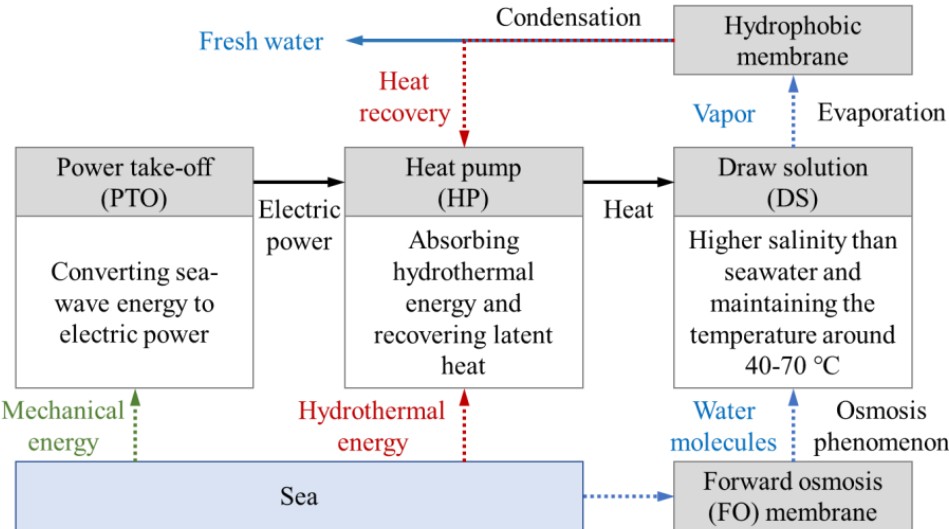

**Figure 1.** Key processes of WavoWater to produce fresh water from seawater.

To secure the process of Figure 1, WavoWater is designed as a cylindrical feature and divided into upper (the outside hollow cylinder) and lower (the inside cylinder) bodies (Figure 2a). There is a space between the upper and the lower bodies, and this gap is filled with air. Therefore, the water of the DS is vaporized as long as the vapor pressure of it is higher than that of the air gap (Figure 2b). The thickness of the air gap increases or decreases according to the interaction between two bodies. The stoppers determine the minimum and maximum clearance of the air gap. As the thickness of the air gap of AGMD increases, the vapor flux decreases rapidly [11,12]. Therefore, the maximum gap thickness of WavoWater was designed to be 10 mm. There is a pipe connecting the air gap and the atmosphere, and the check valve attached to the pipe opens when the pressure inside the gap becomes greater than atmospheric pressure so that condensate and air can be discharged ($P_1$, Figure 2a). On the other hand, when the air gap volume increases due to the relative motion of the upper and lower bodies, the air-gap pressure is maintained at sub-atmospheric pressure with the check valve closed ($P_2$, Figure 2b). Thus, the lowest absolute pressure of the gap is created when the lower body hits the stoppers and the gap thickness is at maximum. The density of the lower body is designed to be higher than those of the upper body and seawater. Thus, the lower body tends to sink while the whole system (WavoWater) is floating. Thus, the relatively different heaving motions of the two bodies provide the gap thickness and pressure changes without an electric power source.

For the system design, Mechanical Room 1 and 2 of the upper body have mechanical devices, such as power take-off (PTO), and parts of an HP including condensing surface (the economizer and evaporator of the HP). The lower body contains the DS, a condenser of the HP for heating, and the MD and FO membranes.

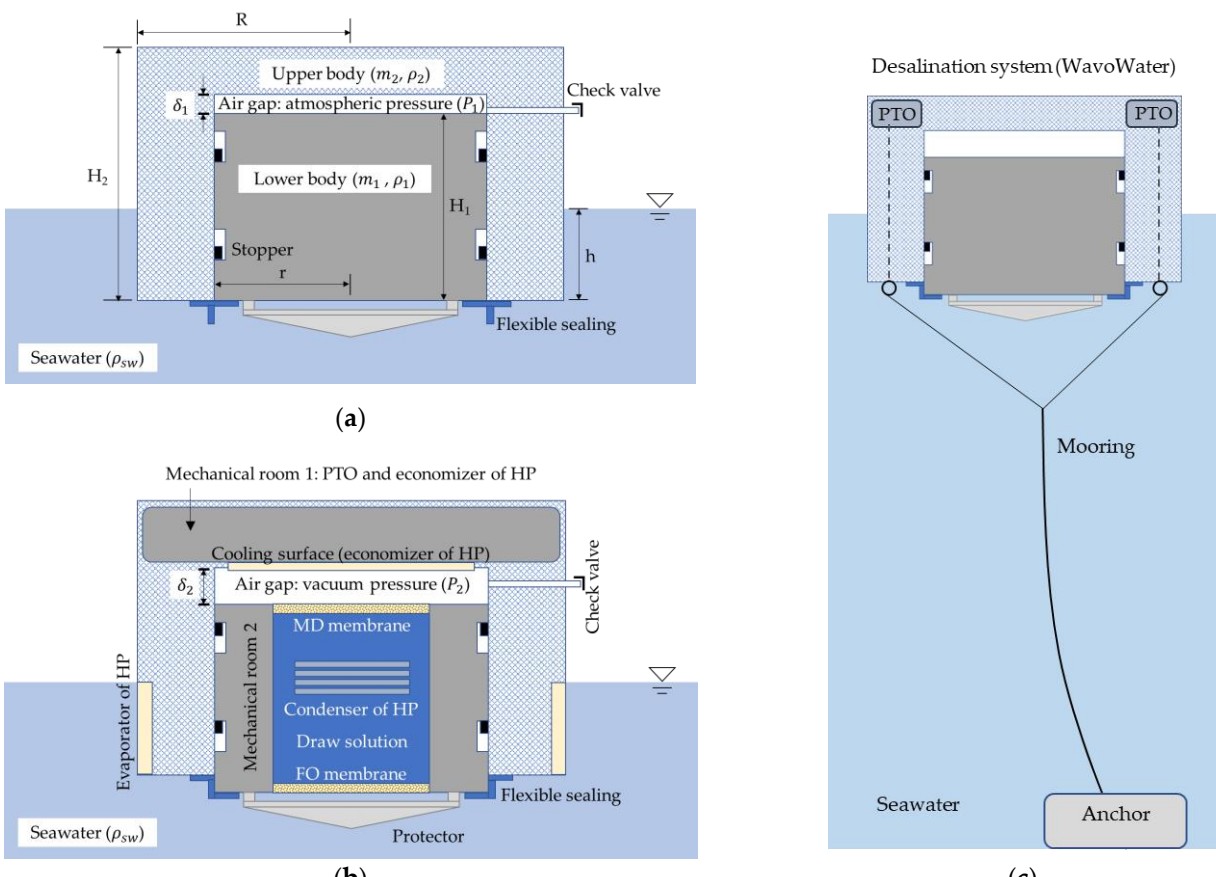

**Figure 2.** The concept of the wave-powered desalination system consisting of two different density bodies ($p_1 > p_2$): (**a**) atmospheric pressure status of the air gap when the upper and the lower body are closest; (**b**) sub-atmospheric condition of the air gap due to the volume expansion caused of the density difference; (**c**) mooring system of WavoWater.

### 2.2. Vacuum-Enhanced Air-Gap Membrane Distillation (AGMD)

Among many desalination technologies, MD is a membrane-based and thermally driven desalination process. The MD has been recently drawing attention because it has great potential to combine renewable heat sources or waste heat, as it can desalinate aqueous solutions by utilizing low-grade waste heat sources. The MD uses a hydrophobic membrane to ensure that only water vapors pass through membrane pores when the vapor pressure difference between the feed and permeate sides is created. The MD is technically classified into four configurations: direct contact membrane distillation (DCMD), air gap membrane distillation (AGMD), vacuum membrane distillation (VMD), and sweeping gas membrane distillation (SGMD) [13]. In the DCMD, an aqueous feed solution and cold water face a hydrophobic membrane in between. Thus, vapors coming from the feed are condensed directly as they move into the permeate side. For the AGMD, a thin air gap exists between a membrane and an internal condensing surface. Therefore, the evaporated water molecules from the surface of the feed solution pass through the membrane and are condensed on the cold surface. On the other hand, a vacuum is applied at the permeate side for the VMD, and the vapor is condensed at an external condenser. The SGMD uses a cold inert or sweeping gas to sweep vapor in a membrane module and condense it at an external condenser [13]. Among these representative configurations, the WavoWater is based on the AGMD as there is an air gap between the MD and the condensing surface. However, since the pressure of the air gap of WavoWater is periodically maintained below atmospheric pressure, it has different characteristics to the general AGMD operated under atmospheric pressure.

In the MD, the vapor flux ($J_{MD}$) through the hydrophobic membrane pores could be explained by Equation (1) [14], and the vapor pressure difference across the membrane ($p_{fw} - p_p$) is regarded as a driving force for the transports of the water vapor.

$$J_{MD} = K_{MD}\left(p_{fw} - p_p\right) \tag{1}$$

where $K_{MD}$ is the mass transfer coeffcient of the hydrophobic porous media, and $p_{fw}$ and $p_p$ are the vapor pressure at the membrane surface on the feed and permeate sides, respectively. Considering Equation (1), increasing the feed solution temperature or decreasing the vapor pressure of the permeate side can improve the overall vapor flux of the system. Therefore, recovering waste heat [15] or utilizing renewable energies, such as solar–thermal [15,16] and geothermal energy [17], can be applied to heat the feed solution of MD. To achieve a low-vapor pressure on the permeate side of MD, the condensation enhancement for inert condenser [18,19] or vacuum pump operation [20,21] could be considered. In addition to these performance-enhancement methods, the removal of non-condensable gases in an air gap could be considered since the non-condensable gases in the gap could hinder the incoming gases to be condensed on a cooling surface [22]. Therefore, auxiliary vacuum pumps were used in AGMD studies [21,23] to eliminate the air gap's negative effect and improve the condensation performance. Regarding the techniques to increase the vapor flux of AGMD, the feed-side solution (DS) of WavoWater could be heated up by the wave-powered HP to create a high vapor pressure, $p_{fw}$. At the same time, the vapor pressure of permeate side (the air gap) of WavoWater is kept lower than that of a typical AGMD operating under atmospheric pressure, as it can create sub-atmospheric pressure at the air gap by heaving motions.

### 2.3. Forward Osmosis Membrane

Without supplying raw water properly from the outside, the salt concentration of DS of WavoWater will gradually increase and reach the limit, at which point distillation does not occur anymore. Therefore, to supply the feed into the DS, WavoWater applies FO membrane at the bottom of the lower body (Figure 2b).

Forward osmosis systems are techniques that utilize the diffusion of water across a selectively permeable membrane from a solution of higher concentration to a solution of lower concentration. Therefore, the DS on the permeation side of the porous membrane becomes the driving force for the osmotic transport of water molecules. In other words, the DS of WavoWater, which remains above the concentration of seawater, can absorb water from the surrounding seawater through the FO membrane. The osmotic pressure differential ($\Delta\pi$) across the membrane is the critical factor to create a mass transfer, rather than hydraulic pressure differential [24]. The water flux of FO can be described by Equation (2) [24]:

$$J_{FO} = K_{FO}(\sigma\Delta\pi - \Delta P) \tag{2}$$

where $K_{FO}$ is the water permeability constant of the FO membrane, $\sigma$ is the reflection coefficient, and $\Delta P$ is the applied pressure.

For this system, we assumed using a higher concentration of sodium chloride than that of seawater as a draw solution and the FO membrane commercially used in hydration bags [24]. Moreover, this study did not consider a salt rejection rate [25], the effect of the concentration polarization [26], and a membrane fouling issue in a FO process.

### 2.4. Heat Pump (HP) System

The role of the HP of WavoWater is to heat the high concentration solution (DS) in the system after recovering the latent heat of the condensate and absorbing the thermal energy of seawater. Considering that the MD system heats the feed to 40–80 °C, a commercial HP system for water heating (HPWH) can be applied. Like the air-conditioning systems, the HPWH uses a vapor-compression refrigeration cycle to deliver the heat absorbed from a low-temperature source to a high-temperature reservoir. According to a litera-

ture review [27], the high-temperature reservoir of present HP technologies can maintain temperatures from about 35 to 125 °C. Among many HPWH systems, several technologies applicable to the WavoWater for providing hot aqueous solutions are summarized in Table 1. Considering the heat recovery from the distilled vapor and seawater thermal energy, the good heating performance using two heat sources at different temperature levels is critical to the WavoWater. In terms of this aspect, Arpagaus et al. [28] demonstrated that their two-stage HP using an open economizer could supply hot water at around 60 °C by absorbing two different heat source levels and achieving a high coefficient of performance (COP). Moreover, According to Bertsch et al. [29], a two-stage HP with an economizer may achieve reasonable efficiency, cost competitiveness, and easy installation.

**Table 1.** Summary of HP technologies applicable to the WavoWater.

| Design High Temp. [°C] | Design Middle and Low Temp. (MT, LT) [°C] | Cycle | Refrigerant | COP | Ref. |
|---|---|---|---|---|---|
| 45 | LT: −8 | Single-stage vapor compression | R290 | - | [30] |
| 55 | LT: 23 | Single-stage vapor compression (water-to-water) | R134a | 2.65 | [31] |
| 55 ± 5 | MT: 30 ± 10; LT: 5 ± 5 | Two-stage vapor compression with an open economizer (water-to-water) | R134a | 4.3 | [28] |
| 45 | MT: 7–11; LT: 5 | Single-stage two-heat source vapor compression with ejector cycle | R134a | 6.9 | [28] |

Considering the DS temperatures, multi-heat sources, and the COPs of HPs, a two-stage vapor compression refrigeration cycle with an economizer was reviewed for the WavoWater. Figure 3 shows the schematic diagram of the two-stage HPWH system for two heat sources at two different temperature levels (water vapor and seawater) and its cycle in the P-h diagram. The cycle consists of two compressors for work input, a condenser as a heat sink at the high-temperature reservoir (draw solution), an open economizer for the first heat source (water vapor), two expansion vales to decrease refrigerant pressures, and an evaporator for the second heat source (seawater). The PTO of the system provides the compressor work.

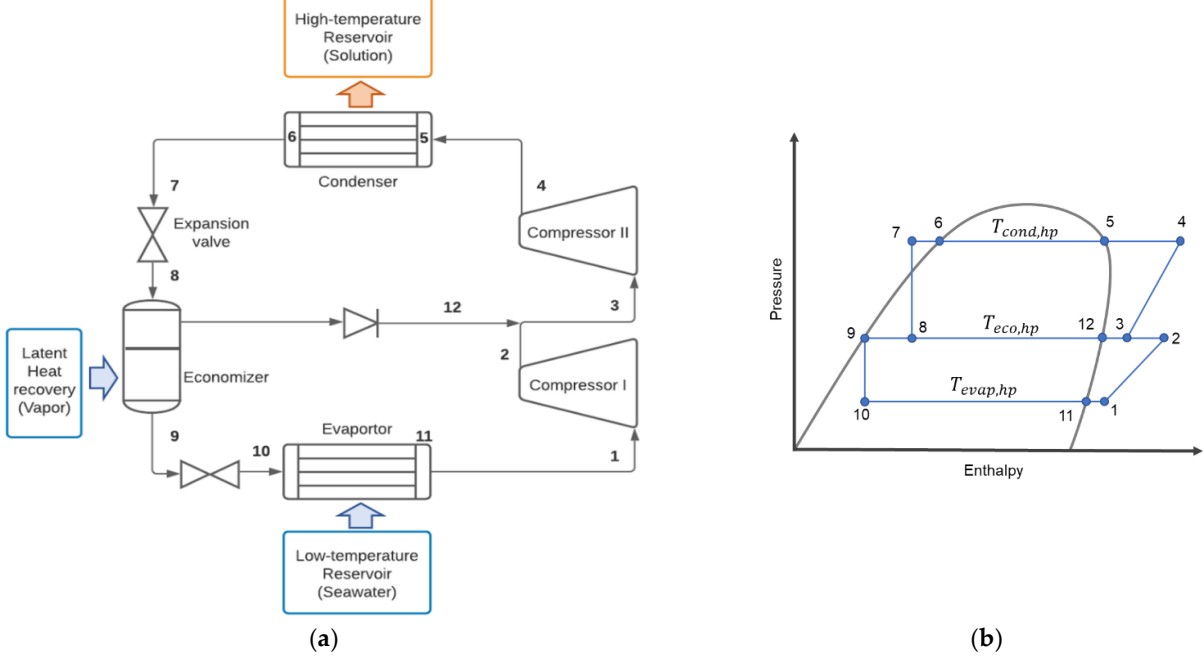

**Figure 3.** Two-stage HP with an economizer [29] for WavoWater: (**a**) Schematic diagram, (**b**) cycle in P-h diagram.

The operating conditions of WavoWater's HP system are summarized in Table 2. The system efficiency ($\eta_{sys}$), considering the heat loss due to the temperature difference between the DS and surrounding temperatures, the energy loss due to the frictions of the PTO systems, and a circulation pump system in the DS, is assumed to be 0.8.

**Table 2.** Input values for the heat pump system.

| Parameter | Value | Description |
|---|---|---|
| Refrigerant (REF$) | R134a | Heat pump refrigerant |
| $T_{sw}$ | 20 [°C] | Seawater temperature |
| $T_1 - T_{11}$ | 3 [K] | Superheating temperature of the evaporator |
| $T_4 - T_5$ | 3 [K] | Superheating temperature of the condenser |
| $T_5$ | $T_{drw} + 3$ | Condenser temperature |
| $T_6 - T_7$ | 2 [K] | Subcooling temperature of the condenser |
| $T_{10}$ | 4 [°C] | Setting inlet temperature of the evaporator |
| $T_{drw}$ | 40, 50, 60 [°C] | Draw solution temperature |
| $T_{eco,hp}$ | 14 [°C] | Economizer surface temperature |
| $\eta_{comp}$ | 0.95 [-] | Compressor motor efficiency |
| $\eta_{isen}$ | 0.9 [-] | Isentropic efficiency of compressor |
| $\eta_{sys}$ | 0.8 [-] | System efficiency including energy losses |
| $P_{drp,evp}$ | 10 [kPa] | Pressure drop at the evaporator |
| $P_{drp,cond}$ | 50 [kPa] | Pressure drop at the condenser |
| $W_{input}$ | 1.4 [kW] | Electric power input from the PTO |

*2.5. Wave Energy Converter (WEC)*

There are numerous methods to generate electric power or mechanical work from oceanic energy. This paper reviews surface wave motions to extract energy to operate the desalination system, WavoWater. A WEC can convert the wave energy directly into electricity to operate the system. The proposed concept was reviewed to supply power to the heat pump system as a single-body point absorber. Among single-body point absorbers, some studies [32,33] applied a flywheel energy storage (FES) device (Figure 4) and demonstrated its superiority in terms of energy harvest and efficiency. Therefore, a WEC combined with a flywheel was considered for the desalination system.

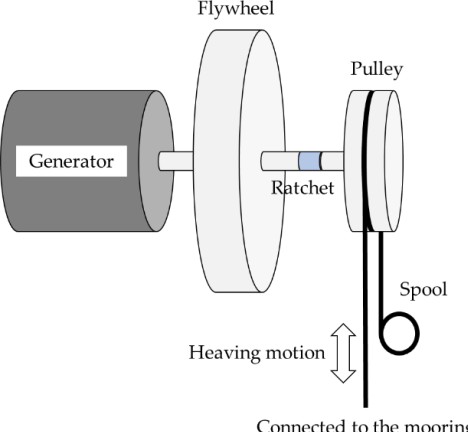

**Figure 4.** A typical flywheel energy storage device and a power take off (PTO) located in the Mechanical Room 1 of Figure 2a.

## 3. WEC Modeling

*3.1. Design Parameter*

The most important parameter that drives the design of a WEC using heaving motions is the natural frequency of the system, as shown in Equation (3). The hydrostatic stiffness of the floating cylindrical body, $k_{sys}$, is proportional to the submerged cross-sectional area

($k_{sys} = \rho_{sw}g\pi r^2$), and the total mass of the system is proportional to the displacement volume of the seawater.

$$\omega = \sqrt{\frac{k_{sys}}{m_{sys}}} = \sqrt{\frac{\rho_{sw}g\pi R^2}{\rho_{sw}\pi R^2 h}} = \sqrt{\frac{g}{h}} \tag{3}$$

The system's total weight should be equilibrant to the buoyancy force of the submerged volume (Equation (4)):

$$(m_1 + m_2)g = \rho_{sw}\cdot\pi R^2\cdot h\cdot g \tag{4}$$

Moreover, considering the hydrostatic equilibrium among the WavoWater, seawater and air pressure, Equation (5) can be obtained.

$$\left(m_1\cdot g - \rho_{sw}\cdot\pi r^2\cdot h\cdot g\right) - \left\{\rho_{sw}\cdot\pi\left(R^2 - r^2\right)\cdot h\cdot g\right\} = (P_1 - P_2)\cdot\pi r^2 \tag{5}$$

From Equations (4)–(6), the mass of the lower body can be calculated by:

$$m_1 = \frac{(P_1 - P_2)\cdot\pi r^2}{2g} \tag{6}$$

As shown in Equation (6), estimating the mass of the lower body, $m_1$, is independent of the density of the surrounding fluid and the immersion height of the body. Once the target vacuum pressure of the air gap and the radius of the cylindrical body were decided, the mass of the lower body could be determined. As we set the final sub-atmospheric pressure and the lower body radius to be 60 kPa and 0.3 m, respectively, the mass of the lower body ($m_1$) was consequently calculated to be 590.9 kg.

To decide the mass of another body ($m_2$), we assumed other parameters as follows: (i) the submerged depth, $h = 0.7H_1$; (ii) the upper body radius, $R = 0.5$ m ; and (iii) the lower body density including mechanical units and a draw solution, $\rho_1 = 1500$ kg·m$^{-3}$. Based on the hydrostatic equations and the assumptions, the design parameters were obtained in Table 3.

**Table 3.** Design parameters for WEC of WavoWater.

| Symbol | Value | Symbol | Value |
|--------|-------|--------|-------|
| $P_1$ | 101 [kPa] | $\rho_{sw}$ | 1024.9 [kg·m$^{-3}$] |
| $P_2$ | 60 [kPa] | $\rho_1$ | 1500 [kg·m$^{-3}$] |
| $h/H_1$ | 0.7 [-] | $\rho_2$ | 200 [kg·m$^{-3}$] |
| $r$ | 0.3 [m] | $m_1$ | 590.85 [kg] |
| $R$ | 0.5 [m] | $m_2$ | 194.13 [kg] |
| $G$ | 9.81 [m·s$^{-2}$] | $H_1$ | 1.39 [m] |
| $H$ | 0.98 [m] | $H_2$ | 1.74 [m] |

*3.2. Power Output Model*

The motion of the system in the wave can be described by Newton's Second Law of Motion:

$$m_{sys}\cdot\ddot{x} = f_d - B\dot{x} - \rho gSx + f_{PTO} \tag{7}$$

where $m_{sys}$ is the total mass of WavoWater and added mass displaced by the submerged part of it; $\ddot{x}$—the vertical acceleration of the floating body; $f_d$—the excitation force the is zero in calm water; $B$—the radiation damping; $\dot{x}$—the vertical velocity; $\rho$—the density of the system; $g$—the acceleration of the gravity; $S$—the cross sectional area of the system; $x$—the vertical motion length; and $f_{PTO}$—the PTO force. The viscous effects are neglected in Equation (7).

The PTO force, $f_{PTO}$, can be rewritten as $-C\dot{x} - k_{sys}x$. $C$ and $k_{sys}$ are the linear damper coefficient and the stiffness of a linear spring, respectively. Assuming a regular wave of frequency ($\omega$) as:

$$\{x, f_d\} = Re(\{X, F_d\}^{iwt}) \tag{8}$$

where $X$ and $F_d$ are complex amplitudes. Based on Equation (7), it can be obtained as:

$$X = \frac{F_d}{-\omega^2 m_{sys} + i\omega(B + C) + \rho g S + k_{sys}} \tag{9}$$

The observed average power over a given time period is estimated as $\overline{P} = \overline{f_d \dot{x}}$. Moreover, the FES inside WavoWater can be modeled by:

$$I\alpha = T_{wire} \cdot r_p - M_{generator} - M_{spool} \tag{10}$$

where $I$ is the inertia of the rotating elements, and $\alpha$ is the angular acceleration. $r_p$ is the radius of the pulley, $M_{generator}$ is the torque of the generator, and $M_{spool}$ is the torque produced by the spool.

### 3.3. Random Wave Generation

Ocean waves are usually random and irregular so that they can be expressed by the energy spectrum over a range of frequencies. Thus, the waves were randomized to give more realistic results from the modeling work, inputting the amplitude and frequency based on the Bretschneider spectrum [34,35]. Figure 5 presents the scatter data of the annual wave probability distribution of significant wave height, $H_{m0}$, and period of wave energy, $T_e$. Most of the significant heights and energy periods are within the range of 0.75–3.75 m and 7.5–12.5 s, respectively, taking over 86% of annual waves. Based upon the annual wave probability distribution of sea in front of the City of Newport, the input parameters for the random wave generation were selected as shown in Table 4.

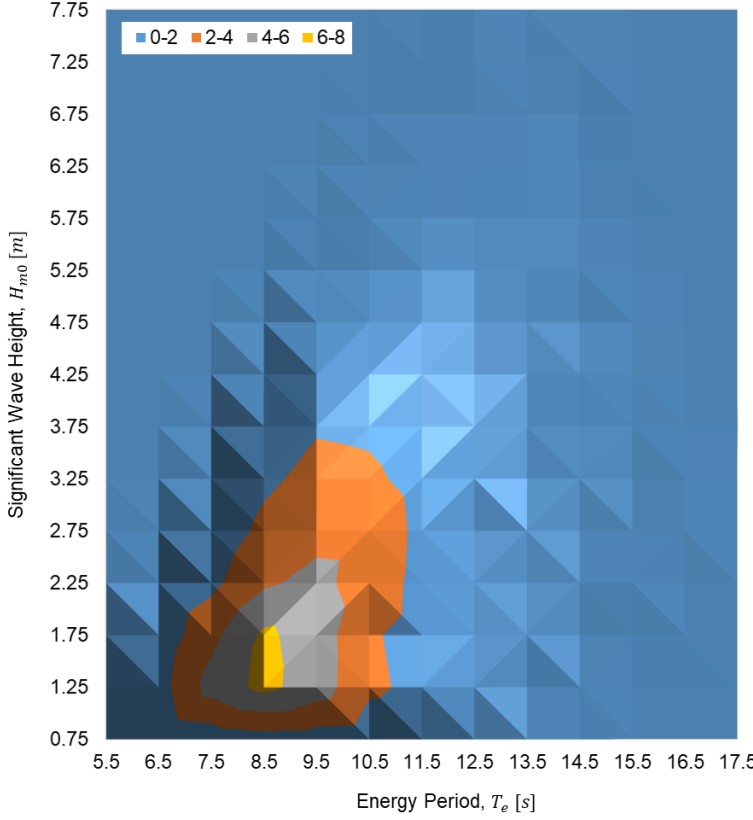

**Figure 5.** Annual wave probability distribution in the coastal waters near the City of Newport, Oregon (Data source: [36]).

**Table 4.** Input values for the random wave generation.

| Symbol | Value | Description | Data Source |
|--------|-------|-------------|-------------|
| $A_{mean}$ | 2.323 [m] | Mean wave amplitude | |
| $A_{s.dev.}$ | 1.128 [m] | Standard deviation of amplitude | [36] |
| $f_{mean}$ | 0.098 [Hz] | Mean wave frequency | |
| $f_{s.dev.}$ | 0.606 [Hz] | Standard deviation of frequency | |

The wave frequency and amplitude of randomized waves were generated (Figure 6) using MATLAB and used as input values of the main simulation program to estimate the power of the system. The mean wave period, $T_{mean}$, is the mean of all wave periods in a time series representing a specific sea state, and the mean wave frequency is the reciprocal of the mean wave period ($f_{mean} = 1/T_{mean}$).

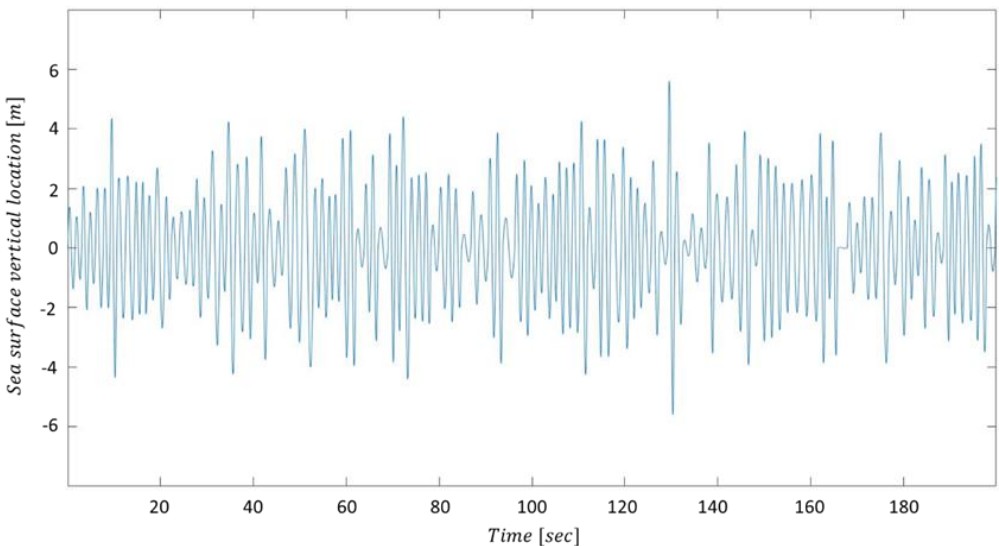

**Figure 6.** Time-series of surface elevation of the randomized waves.

*3.4. Power Output Analysis*

Considering the negative effect of the back-torque, Helkin (2011) [37] showed that the FES-based WEC system could increase the average power output by controlling the generator load. The lower and upper threshold values were examined to optimize the FES-WEC system—i.e., the lower threshold value is the rotor velocity (revolutions per minute, RPM) to be disengaged between the FES and generator, and the upper threshold value is the value to be engaged. By using Helkin's model and method in MATLAB, we analyzed the average power output of WavoWater. As the result of 200 cycles of iteration, the mean value of average power was estimated, as Figure 7 shows. It should be noted that the point equaled to zero represents the status without load control. Moreover, the load control status remains the same as its previous state until the RPM falls below the lower threshold value or increases above the upper threshold. The upper threshold value is always equal to or above the lower threshold parameter. From Figure 7, the maximum average power output (1.45 kW) is obtained when the FES is controlled within the optimal lower and upper threshold values, 28 and 76 RPM, respectively.

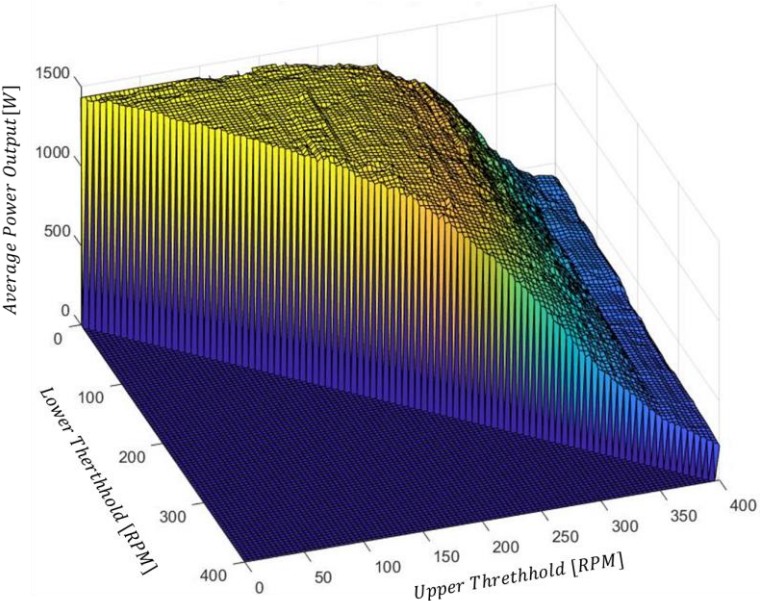

**Figure 7.** Average power output of WavoWater according to the lower and upper threshold values.

## 4. Water Production Estimation

The wave power absorbed from the PTO is converted into electrical energy and drives HP of which the condenser heats DS to distill water vapor from it. Since the air in the gap between the upper and lower body has a check valve at the outlet end, assuming that the weight of the check valve is negligible, the pressure inside the gap can be considered to remain sub-atmospheric pressure while it is closed. Furthermore, the pressure of the gap may be estimated by the assuming it as a polytropic process as it includes heat transfer while the compression–expansion process is repeated by the relative motion of the upper and lower parts of WavoWater until the check valve is opened (Equation (11)) [38]:

$$PV^n = constant \tag{11}$$

The value of the polytropic exponent, n, can be regarded as 1.4 for the atmospheric and sub-atmospheric pressure states [26]. For the WavoWater operation, we assumed that the minimum and the maximum gap-thickness were 6.4 and 9.6 mm, respectively. The air gap pressure was at the atmospheric pressure with minimum gap thickness. As the gap thickness increases, the pressure of the air gap decreases, as shown in Figure 8. Therefore, the lowest testing pressure of the air gap was set at around 60 kPa.

A model of AGMD's vapor flux with the air gap maintained under the sub-atmospheric pressure is essential for analyzing the performance of WavoWater. Since it is hard to find previous literature on AGMD distillation experiments in sub-atmospheric pressure with the range of 60–100 kPa, a simple experimental device was designed and built to observe the change in the vapor flux of the AGMD according to the air-gap pressure change (Figure 9). As there is a study result showing that the effect of vapor flux change in AGMD according to the feed flow rate is insignificant [39], the experiment was designed and performed by giving a minimum flow rate of the saline feed of the AGMD module to satisfy the condition of maintaining a constant temperature of the draw solution in the lower body. A flat-sheet hydrophobic membrane manufactured by Millipore was used for the test. Its main characteristics, as specified by the manufacturer, are as follows: (1) material—polytetrafluoroethylene (PTFE) polymer; (2) pore size of the membrane—0.22 μm; (3) thickness is 150 μm; (4) porosity is 85%; and (5) diameter—142 mm.

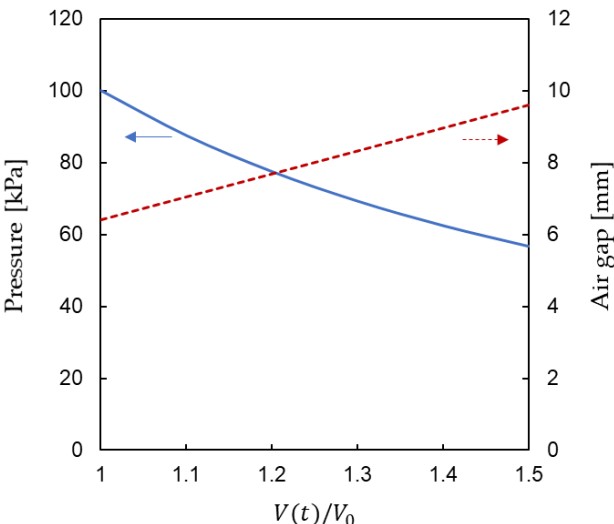

**Figure 8.** The change of the pressure and the gap thickness according to the volume change of the gap between the upper and the lower bodies of WavoWater.

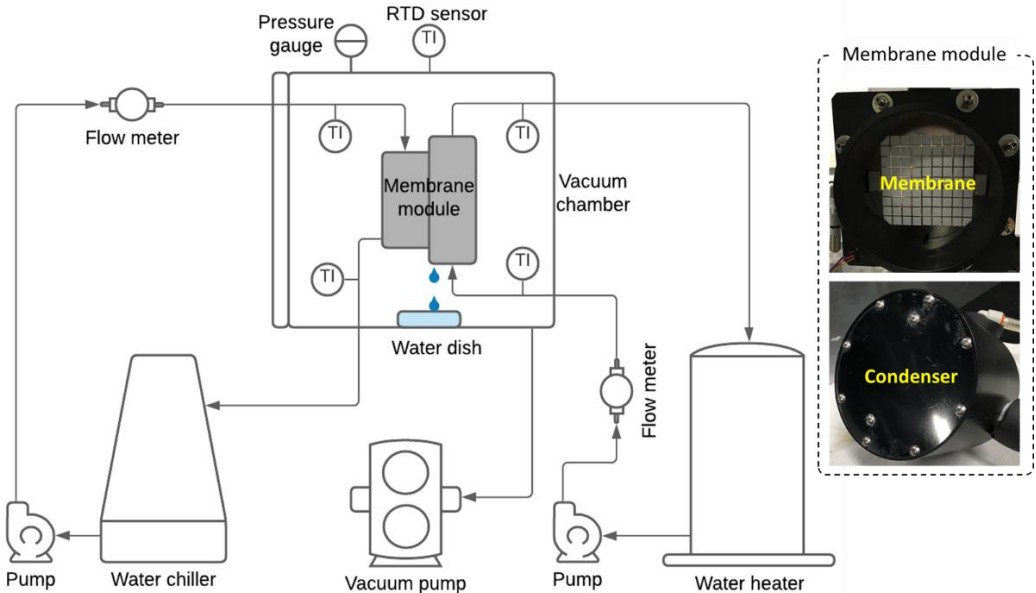

**Figure 9.** Schematic diagram of the bench-scale test facility for vacuum-applied AGMD.

Considering the osmotic pressure difference between the DS and seawater around WavoWater, the salinity of the feed was set to 40 g·L$^{-1}$. During the test, we ignored the concentration polarization effect [24] on the FO and MD membranes. The feed water temperature was measured constantly at the inlet and outlet of the membrane module with an accuracy of ±0.025 °C. FLUKE's RTD probes were installed and connected to Nation Instruments' data acquisition (DAQ) hardware. The feed water was heated by a 3 M polyimide film heater in the hot water tank. Setra's pressure transducer having the range of 500–1100 mbar with precision ± 0.02% was used to measure the air pressure of the chamber. The condenser surface to condense water vapor was made of stainless steel. We installed a diverter flap controlled by a stepper motor below the drain hole of the module to collect the produced water at the dish only for the measuring period.

We tested thewater generation rate under atmospheric and sub-atmospheric pressure (60–101 kPa). The feed water and the coolant flow rates were fixed at 10.9 mL·s$^{-1}$ and 16.7 mL·s$^{-1}$, respectively. We ignored the quantity of the evaporated and condensed water at the water surface of the dish. At first, we measured the produced water flux

($J_m$) coming out of the AGMD module for each different air-gap pressure ($P_{ag}$) and feed water temperature (40, 50, and 60 °C) with the fixed condenser coolant temperature (10.4 °C) (Figure 10). The water flux of the AGMD decreased with the air-gap pressure at the constant feed liquid temperature and the feed side temperature increased. These results can be explained as follows: the vapor pressure difference ($p_{fw} - p_p$) in Equation (1) increases as either the vapor pressure ($p_p$) of the permeate side decreases along with the total air pressure ($P_{ag}$) lowering or the vapor pressure ($p_{fw}$) of the feed side increasing according to the increased temperature.

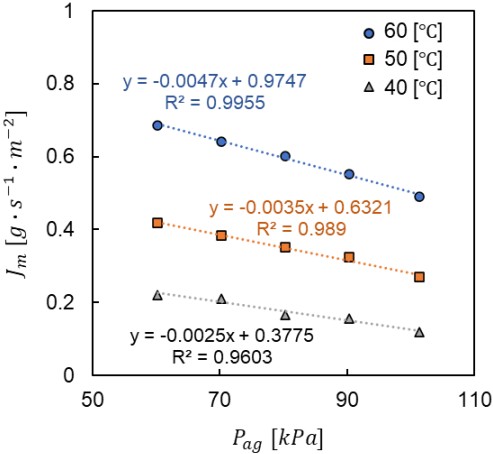

**Figure 10.** Produced water flux according to the air-gap pressure and feed and coolant water temperatures.

## 5. Results and Discussion

Water production capacity should be considered as one of the most important factors to ensure the reliability of grid-independent desalination technology as an alternative water supply system. Therefore, the water production capacity of WavoWater was evaluated according to the change in the operating temperature of DS as shown in Figure 11. The pressure change of the air gap formed by the relative motion of the upper and lower parts of the WavoWater was simulated based on the random wave motion in Figure 6. It was shown that the higher the operating temperature of DS, the higher the production quantity per unit of WavoWater. When the temperature of DS was increased from 40 °C to 60 °C, the production capacity was expected to increase rapidly to about 244%, while an increase of 94% was expected at 50 °C operation.

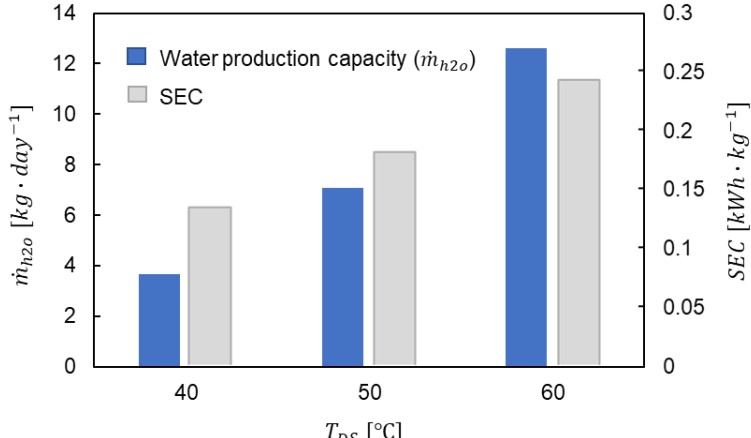

**Figure 11.** Water production capacities and SECs of a single WavoWater system according to the DS temperature ($T_{DS}$).

As an energy-independent desalination device, WavoWater needs to be designed in consideration of the amount of energy required according to the production quantity of fresh water. The specific energy consumption (*SEC*) is defined as the amount of energy ($P_{in}$) supplied to produce a unit mass ($\dot{m}_{pro}$) of the product:

$$SEC = \frac{P_{in}}{\dot{m}_{pro}} \tag{12}$$

This performance criterion in this study is used to evaluate the electricity utilization performance of a desalination system level.

The wave energy consumed to produce pure water per unit mass is shown in Figure 11. It can be seen that the required wave energy increases as the operating set temperature of DS increases. These results may explain with the reason that HP's COP is higher as the temperature level of the high temperature reservoir of the HP system is lower. In other words, maintaining the DS at a high temperature requires higher energy conversion from WEC to produce a unit quantity of fresh water from WavoWater.

Wave energy required for WavoWater's operation was analyzed as shown in Figure 12. When the DS operating temperature range was 40 °C to 60 °C, the energy required to power the HP installed inside 1 m-diameter WavoWater was between 0.5 and 3.1 kWh. Considering the wave energy that can be supplied for 24 h, it can be operated not only at the coast of Oregon, as considered in this study, but also in other regions with much lower wave energy.

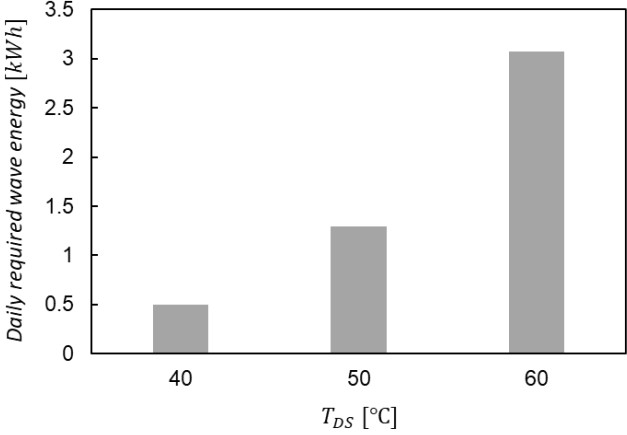

**Figure 12.** Daily required wave energy for the WavoWater operation according to the DS temperature.

In addition, applying the FO system to WavoWater can have a positive effect in terms of environmental impact. Figure 13 describes a conventional desalination system and WavoWater with all energy and material streams, such as seawater, freshwater, and brine. General desalination plants except for zero liquid discharge desalination (ZLDD) discharge brine into the sea after separating freshwater from feed seawater (Figure 13a). As reported by previous studies [32,36], the discharged high-concentrated solution may cause environmental issues and disturb the ecosystem equilibrium. Thus, a ZLDD system combined with a solar pond or evaporator is applied to minimize environmental impacts even though it requires more initial and operating costs. Otherwise, the WavoWater can skip the brine discharge since FO is applied. Only water molecules can pass through a FO membrane by a natural osmotic phenomenon and be separated by the heat released from a heat pump. Therefore, it can produce freshwater without generating concentrated brine (Figure 13b).

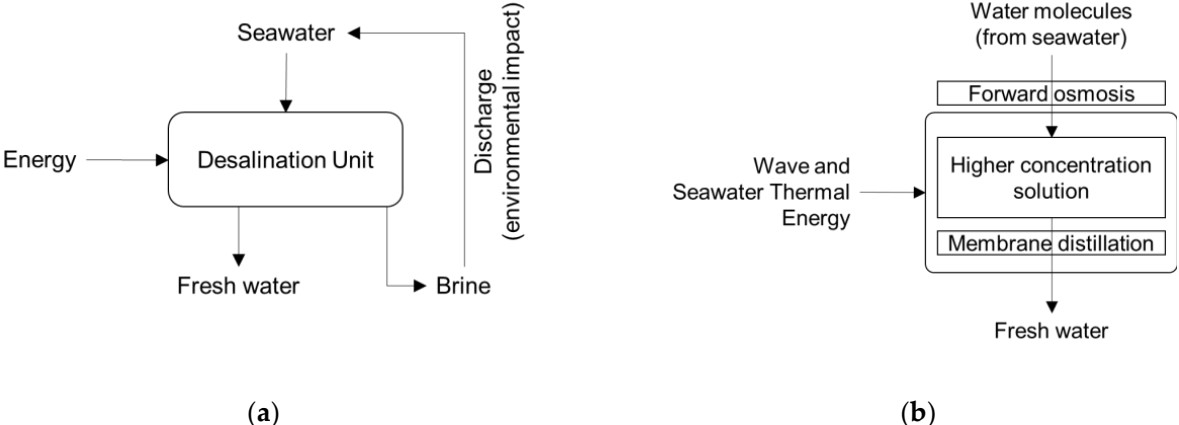

**Figure 13.** Comparison between (**a**) a conventional desalination system and (**b**) WavoWater.

## 6. Conclusions

Many studies have shown that wave power can be transformed to produce fresh water by combining a desalination system and WEC. Unlike previous studies, this paper proposes a new hybrid system combining the HP system using wave power and the AGMD desalination process. In addition, the new wave-powered desalination system, WavoWater, is equipped with an FO membrane and is designed to produce fresh water without discharging concentrated salt water. HPWH has been reviewed as a device for supplying a heat source for AGMD process, and the inside of the air gap is designed to create a vacuum by relative motion between the upper and lower parts of WavoWater. WavoWater, which can produce about 12.6 kg of fresh water per day at a DS operating temperature of 60 °C, has the following advantages:

(1) WavoWater can produce freshwater as a desalination system without the discharge of high-concentration solutions.
(2) WavoWater can supply drinking water to the areas that lack grid electricity by using wave power.
(3) WavoWater can be demonstrated as a small-scale modular system. Consequently, it is easy to transport and deploy.
(4) The desalination process of WavoWater is much simpler than any other desalination system by skipping pumping systems and a regular backwash process for filtration units. Thus, its maintenance costs can be low, and it may not require a high level of technical knowledge from a customer.

The WavoWater, the stand-alone wave-powered desalination system, can be helpful to many remote or disaster areas with unreliable power grids. Furthermore, this portable and the easy-deployed system is expected to supply drinking water everywhere, including unique circumstances such as disaster response and remote coastal communities.

**Author Contributions:** Conceptualization, G.S.K.; methodology, G.S.K.; experimental design and setup, G.S.K.; experiment, G.S.K.; numerical modeling, G.S.K.; validation, G.S.K.; formal analysis, G.S.K.; investigation, G.S.K.; resources, G.S.K.; data curation, G.S.K.; writing—original draft preparation, G.S.K.; writing—review and editing, Y.H.; visualization, G.S.K.; supervision, Y.H.; project administration, G.S.K.; funding acquisition, G.S.K. All authors have read and agreed to the published version of the manuscript.

**Funding:** This research was funded as Link Ocean Engineering & Instrumentation Ph.D. Fellowship Program by the Link Foundation, July 2020–June 2021.

**Institutional Review Board Statement:** Not applicable.

**Informed Consent Statement:** Not applicable.

**Data Availability Statement:** The data presented in this study are available on request from the corresponding author.

**Conflicts of Interest:** The authors declare no conflict of interest that may be perceived as inappropriately influencing the representation or interpretation of reported research results. Moreover, the funders had no role in the design of the study; in the collection, analyses, or interpretation of data; in the writing of the manuscript, or in the decision to publish the results.

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
