# Peer review of "Wave-Powered and Zero-Discharging Membrane-Distillation Desalination System: Conceptual Design and Analysis"

_water, doi:10.3390/w14121897_

Round 1

Reviewer 1 Report

This manuscript presents a novel conceptual design of a floating desalination system driven by wave power. It is an interesting idea to apply the membrane distillation system in near-island regions. The potential readers will also find this manuscript interesting. To improve the quality of this manuscript, here are some comments and suggestions:

(1). The section of results and discussion should be described with more details on the analysis and potential performance of the MD system.

(2).  The section of conclusions should give some highlights of the floating MD system and its potential performance with specific data. 

(3).  In Eqn.(1),  ??? is not the permeability of aporous membrane media, but the mass transfer coefficient of a MD system, which can be considered as constant under common MD conditions. 

Author Response

Thank you for your comments. Please see the attached reply.

Reviewer 2 Report

The authors proposed a new conceptual design of wave-powered MD-FO system. It will be better to explain the difficulty on wave power-based MD systems in Introduction. 

Author Response

(The authors gave the same response as above.)

Reviewer 3 Report

Seawater transfers to fresh water is an important topic in the world. This paper introduces a new system named WavoWater achieve great performance. It is very interesting, while some minor problems should be concerned.

1.     Some important results should be added in the Abstract.

2.     Some figures are not clear, which should be improved.

3.     How about the cost of this new system? If it is profit or loss?

Author Response

(The authors gave the same response as above.)

Reviewer 4 Report

The purpose of the article is to descirbe a new wave-powered MD system for desalination of seawater to produce fresh water. Overall, the article is well written and easy to read. I recommend that the article be published after have addressed the following issues.

1. Abstract is very generic. Please provide a summary of key results such as the energy generated, AGMD system capacity, flux etc.

2. There is no description of the MD and the FO membranes used in this study. Please provide the source and properties of the membranes.

3. As such, there is no experimental section. Please include the detailed description of materials and methods used in this study.

4. Figure 11. The y-axis units are not correct. The SEC is reported as kWh/Kg and not the way it is reported. Please correct.

5. Please revise the references according to the journal format. 

Author Response

(The authors gave the same response as above.)
